# Implementation and Parameter Analysis of the Knock Phenomenon of a Marine Dual-Fuel Engine Based on a Two-Zone Combustion Model

**Fang-kun Zou** [1], **Hong Zeng** [1,*], **Huai-yu Wang** [2], **Xin-xin Wang** [1] and **Zhao-xin Xu** [1]

1   Marine Engineering College, Dalian Maritime University, Dalian 116026, China;
    zfk86902769@163.com (F.-k.Z.); wxx960405@163.com (X.-x.W.); a15734105737@163.com (Z.-x.X.)
2   School of Mechanical Engineering, Beijing Institute of Technology, Beijing 100081, China; nepu2017@126.com
*   Correspondence: zenghong@dlmu.edu.cn

**Abstract:** The stable working window of a dual-fuel engine is narrow, and it is prone to knock during operation. The occurrence of knock limits the load and torque output of a dual-fuel engine, and even causes engine damage in severe cases. The existing volumetric model of marine dual-fuel engine has little research on the related problems of knock simulation. In order to analyze the causes of knock phenomenon and the influence of operating parameter changes on knock, under the Matlab/Simulink simulation environment, a quasi-dimensional model was established with MAN 8L51/60DF dual-fuel engine as the prototype, and the model was calibrated using the bench data. The knock intensity index coefficient (KI) was used as the evaluation index of knock intensity. Three parameters, the intake air temperature, compression ratio, and natural gas intake, were selected as variables to simulate the engine. According to the analysis of the simulation results, the influence of the parameter changes on the occurrence of engine knock phenomenon and knock intensity could be further studied. The results showed that the combination of the KI model and the quasi-dimensional model could effectively and accurately predict the engine performance and knock trend. The change of gas inlet quality, compression ratio, and inlet temperature could promote the occurrence of detonation, the engine knock could be avoided by controlling the intake air temperature below 336 K, compression ratio not exceeding 15 or the intake volume of natural gas per cycle not exceeding 11.25 g/cycle.

**Keywords:** dual-fuel engine; quasi-dimensional model; knock simulation

## 1. Introduction

With the aggravation of global energy shortage and the worsening of environmental problems, pollutant emission has become an important reason for the deterioration of the global atmospheric environment. Shipping industry is one of the most important modes of transportation in international trade, the nitrogen oxides ($NO_x$) emission is the main pollutant emission from ships, and the air pollution caused by its emission already attracts the attention of the international community [1,2].

In order to prevent the global atmospheric environment and marine environment from deteriorating, the International Maritime Organization (IMO) issued new and more stringent standards and revised the previous rules, hoping to reduce the environmental pollution caused by ship emissions. With the shortage of traditional oil resources and increasingly more stringent emission regulations issued by IMO [3], natural gas has the advantages of abundant reserves, high energy density, and green environmental protection, as compared to traditional fuels. Dual-fuel engines using natural gas as the main fuel developed rapidly and are gradually applied in the field of ship power plant.

Although the dual-fuel engine has obvious advantages in emissions and other aspects, the pre-mixed natural gas dual-fuel engine will produce knock phenomenon under the

conditions of excessive load and high temperature in the combustion chamber. When the knock is slight, the engine power can be slightly increased; but when the knock is strong, not only will the engine power output be reduced, but the fuel consumption will be increased and emissions will worsen. At the same time, due to the high temperature and high pressure in the cylinder, severe knocks might even cause damage to the mechanical parts of the engine, as shown in Figure 1, which seriously affects the safety of engine operation [4]. Therefore, it is of theoretical significance and practical value to study the knock combustion characteristics of a dual-fuel engine and the influence of various factors on knocks.

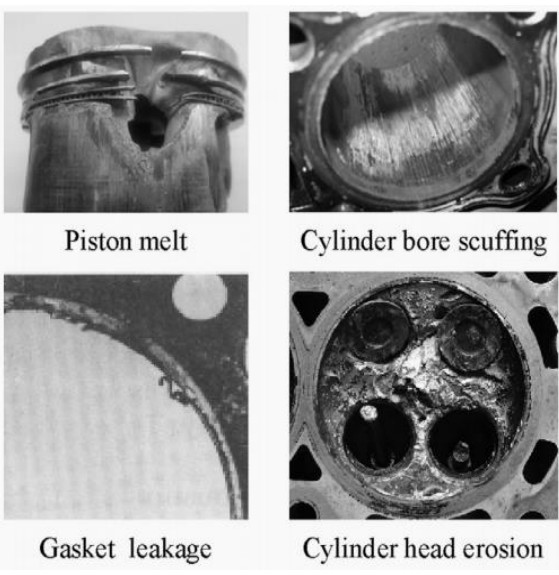

**Figure 1.** Examples of the knocking damage.

There are two main types of detection methods for engine knock phenomenon—one is based on direct measurement methods, such as enhanced charge coupled detector camera and laser-induced fluorescence imaging method [5], the other is an experiment-based method based on indirect measurements [6], such as cylinder pressure analysis, cylinder block vibration, exhaust temperature, etc. In this paper, the primary goal of dual-fuel engine modeling is the knock phenomenon simulation. At present, the knock prediction model mainly includes zero-dimensional model, quasi-dimensional model, and three-dimensional model; different models have advantages and disadvantages in terms of calculation time and accuracy [7].

Abdelaal et al. [8] studied the effects of oxygen on engine performance, emissions and knock tendency, under different load conditions. Wang et al. [9] analyzed the effects of pre-injection on the combustion, emissions, and knock in dual-fuel engine by using 3D Computational Fluid Dynamics (CFD) software. The results showed that the pre-injection strategy was effective in reducing emissions, improving the fuel economy and reducing knock tendencies. To achieve more efficient and robust combustion, Shi et al. [10] split direct-injected hydrogen, as a new injection strategy for rotary engines was used to clarify the influences of various injection parameters, including hydrogen amount for each pulse, the dwell between pulses, and secondary injection pulse-width, on combustion performance and emission formation by CFD modeling. Ma et al. [11] used a self-developed knock prediction model to simulate the performance and knock phenomenon of natural gas engines, and analyzed the numerical results obtained. Krenn et al. [12] proposed a new method for modeling large-scale dual-fuel engines, combining sub-models and thermodynamic dual-zone models, making it possible to determine all required thermodynamic parameters in the combustion chamber and to calculate the mass flow rate from the unburned zone into the combustion zone. The model is more accurate and takes into account the influence of diesel injection. Xiang et al. [13] established a two-zone model to predict

the knock performance and NO emissions of natural gas engines, which was validated with test data from SI natural gas engines. Selim [14] conducted research on dual-fuel engines of three gas fuel combinations of diesel and liquefied petroleum gas, pure methane and compressed natural gas mixture, measured and analyzed the data of combustion noise, knock, and ignition limit. Bares et al. [15] proposed a new knock model by combining the knock integral model based on the in-cylinder temperature and the external noise disturbing the cylinder temperature. The spontaneous combustion of the end gas was modeled by a function similar to Arrhenius. The knock probability was estimated by propagating a virtual error probability distribution. In summary, the quasi-dimensional model had sufficient accuracy in aspects of predicting performance, $NO_x$ emissions, and knock performance. However, most researchers focused their studies on flame propagation processes and applied complex reaction mechanisms to obtain emission concentrations or perform burst simulations, which increases the complexity and computational cost of the models, to a certain extent [16,17].

In order to reduce the damage to the engine and save research time and development costs, the main goal of the simulation modeling of dual-fuel engine in this paper was to simulate the knock phenomenon. A quasi-dimensional model was established on the basis of MAN 8L51/60DF dual-fuel engine as the prototype. To ensure the reliability of the simulation analysis, the model was verified by bench test data. The validity of the established two-zone model was verified. The simulation results were good. On the basis of the simulation model, the influence of intake temperature, compression ratio, and natural gas intake on the knock phenomenon were further studied by changing the engine operating parameters, so as to analyze the causes of the knock phenomenon and the influence of the parameters operating on it.

## 2. Model Description

### 2.1. Two-Zone Model

As shown in Figure 2, the two-zone model is the simplest quasi-dimensional model. It divides the combustion chamber into two-zero dimensional zones, the burned region and the unburned region, by the flame front. The upper half was the burned zone, and the lower half was the unburned zone. The relationship between the two zones was established through the basic equations. The assumptions for the mass and energy exchange between the two zones were as follows:

(1)  The working medium in the cylinder was ideal gas. The mixed gas in each zone was fully mixed and evenly distributed in the cylinder at any instant.

(2)  After pilot fuel injection, evaporation and combustion proceeded instantaneously. The gas components in the burned zone were pilot diesel, natural gas, air, and chemical reaction gas, while those in the unburned zone were natural gas, air, and chemical reaction gas.

(3)  The combustion products of natural gas and air only included $CO_2$, $H_2O$, $O_2$, $N_2$, and Ar. The gas properties of the gas in the cylinder, such as specific heat capacity c, specific internal energy u, and specific heat h, were only related to the mass fraction g and temperature T of the gas component, while the gas constant was only related to g.

(4)  The temperature of the piston head, cylinder wall, and cylinder head was constant and did not change with the crankshaft angle. Since the combustion chamber was divided into two zones—the burned zone and the unburned zone—the burned zone was in contact with the cylinder head and part of the cylinder liner, so the heat loss of the burned zone only included the cylinder head and part of the cylinder liner. The heat loss of the unburned zone only included the cylinder liner (the unburned zone) and the piston crown.

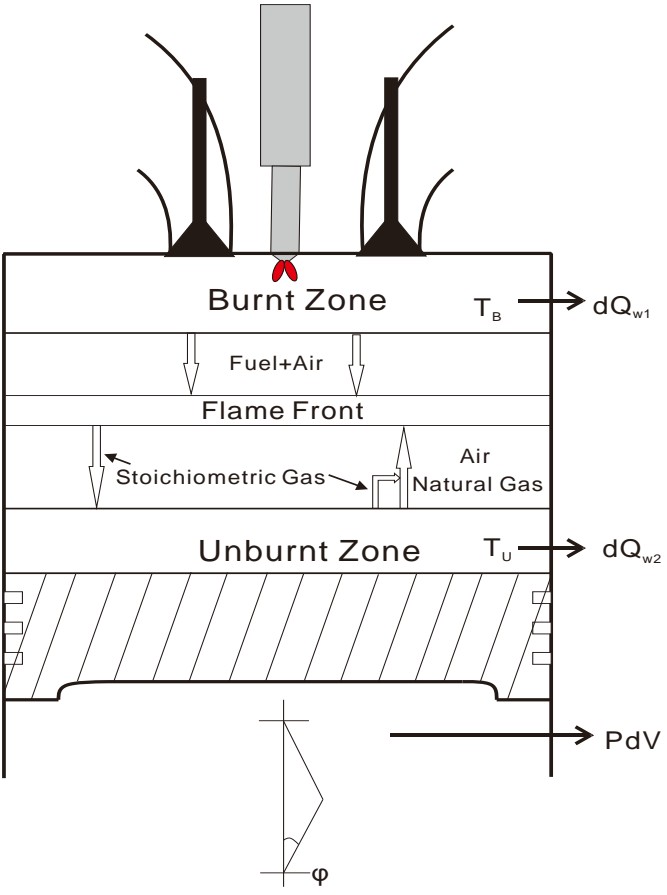

**Figure 2.** Schematic diagram of the two-zone model.

*2.2. Gas Properties*

It can be known from the assumption that the working fluid in the cylinder was an ideal gas. For an ideal gas, the thermodynamic parameters of a single working fluid were only related to temperature. Under setting conditions, the specific heat capacity, specific enthalpy, and specific entropy of a single working fluid in the cylinder could be described by National Aeronautics and Space Administration(NASA) polynomials for thermodynamic properties, such as Equations (1)–(3):

$$\frac{c_{pk}}{R} = a_{1k} + a_{2k}T + a_{3k}T^2 + a_{4k}T^3 + a_{5k}T^4 \tag{1}$$

$$\frac{H_k}{R} = a_{1k} + \frac{a_{2k}}{2}T + \frac{a_{3k}}{3}T^2 + \frac{a_{4k}}{4}T^3 + \frac{a_{5k}}{5}T^4 + \frac{a_{6k}}{T} \tag{2}$$

$$\frac{S_k}{R} = a_{1k}\ln T + a_{2k}T + \frac{a_{3k}}{2}T^2 + \frac{a_{4k}}{3}T^3 + \frac{a_{5k}}{4}T^4 + a_{7k} \tag{3}$$

Table 1 shows the values of the fitting parameters in the temperature range of 300–1000 K and 1000–5000 K. The working fluid in the engine was a mixed gas. A mixture of multiple gas components could be weighted by the weight of a single substance:

$$u = \sum_i g_i u_i \tag{4}$$

$$h = \sum_i g_i h_i \tag{5}$$

$$c = \sum_i g_i c_i \tag{6}$$

$g_i$ is the mass component of the *i*-th gas.

**Table 1.** Fitting parameters.

| | a$_1$ | a$_2$ | a$_3$ | a$_4$ | a$_5$ | a$_6$ | T (K) |
|---|---|---|---|---|---|---|---|
| N$_2$ | 3.298677 | $0.14082404 \times 10^{-2}$ | $-0.03963222 \times 10^{-4}$ | $0.05641515 \times 10^{-7}$ | $-0.02444854 \times 10^{-10}$ | $-1020.8999$ | 300–1000 |
| | 2.926640 | $0.14879768 \times 10^{-2}$ | $-0.05684760 \times 10^{-5}$ | $0.10097038 \times 10^{-9}$ | $-0.06753351 \times 10^{-13}$ | $-922.7977$ | 1000–5000 |
| O$_2$ | 3.212936 | $0.11274864 \times 10^{-2}$ | $-0.05756150 \times 10^{-5}$ | $0.13138773 \times 10^{-8}$ | $-0.08768554 \times 10^{-11}$ | $-1005.2490$ | 300–1000 |
| | 3.697578 | $0.06135197 \times 10^{-2}$ | $-0.12588420 \times 10^{-6}$ | $0.01775281 \times 10^{-9}$ | $-0.11364354 \times 10^{-14}$ | $-1233.9301$ | 1000–5000 |
| Ar | 2.500000 | 0.00000000 | 0.00000000 | 0.00000000 | 0.00000000 | $-745.3750$ | 300–5000 |
| CO$_2$ | 2.275724 | $0.09922072 \times 10^{-1}$ | $-0.10409113 \times 10^{-4}$ | $0.06866686 \times 10^{-7}$ | $-0.02117280 \times 10^{-10}$ | $-48,373.14$ | 300–1000 |
| | 4.453623 | $0.03140168 \times 10^{-1}$ | $-0.12784105 \times 10^{-5}$ | $0.02393996 \times 10^{-8}$ | $-0.16690333 \times 10^{-13}$ | $-48,966.96$ | 1000–5000 |
| H$_2$O | 3.386842 | $0.03474982 \times 10^{-1}$ | $-0.06354696 \times 10^{-4}$ | $0.06968581 \times 10^{-7}$ | $-0.02506588 \times 10^{-10}$ | $-30,208.11$ | 300–1000 |
| | 2.672145 | $0.03056293 \times 10^{-1}$ | $-0.08730260 \times 10^{-5}$ | $0.12009964 \times 10^{-9}$ | $-0.06391618 \times 10^{-13}$ | $-29,899.21$ | 1000–5000 |
| CH$_4$ | 0.7787415 | 0.01747668 | $-0.02783409 \times 10^{-3}$ | $0.03049708 \times 10^{-6}$ | $-0.12239307 \times 10^{-10}$ | $-9825.229$ | 300–1000 |
| | 1.683478 | $0.10237236 \times 10^{-1}$ | $-0.03875128 \times 10^{-4}$ | $0.06785585 \times 10^{-8}$ | $-0.04503423 \times 10^{-12}$ | $-10,080.787$ | 1000–5000 |

### 2.3. Two-Zone Instantaneous Work Mass Model

The in-cylinder combustion chamber was considered to be a closed container in the simulation model, ignoring the leakage loss of the mass in the cylinder. In the in-cylinder combustion chemical reaction, the sum of each substance participating in the reaction was equal to the sum of each substance produced after the reaction satisfied the mass conservation equation.

$$\frac{dm_B}{d\phi} = \frac{dm_f}{d\phi} + \frac{dm_g^{ent}}{d\phi} - \frac{dm_{sg}^{exit}}{d\phi} \tag{7}$$

$$\frac{dm_U}{d\phi} = \frac{dm_{sg}^{out}}{d\phi} - \frac{dm_g^{in}}{d\phi} \tag{8}$$

Subscript $B$ and $U$ were used to describe the parameters related to the burned zone and unburned zone, respectively. In the above equation, $dm_B/d\phi$ is the instantaneous mass change rate in the burned area, $dm_U/d\phi$ is the instantaneous mass change rate in the unburned area, $dm_f/d\phi$ is the rate of mass change of evaporated fuel in the burned zone, $dm_g^{ent}/d\phi$ is the mass rate that transferred from the unburned zone to the burned zone, including natural gas, air, and chemical reaction gases; $dm_{sg}^{exit}/d\phi$ is the chemical reaction gas transferred from the burned zone to the unburned zone.

### 2.4. Two-Zone Transient Temperature Model

According to the above assumptions, the sub model of each region is zero-dimensional. Equation (9) is derived from the first law of thermodynamics, by separating the temperature terms:

$$\frac{dT_B}{d\phi} = \left[ \frac{dm_g^{ent}}{d\phi} \left( h_g^{ent} - u_{g,1} \right) - \frac{dm_{sg}^{exit}}{d\phi} \left( h_{sg}^{exit} - u_{sg,1} \right) + \frac{dQ_f}{d\phi} - \frac{dQ_{W1}}{d\phi} - \frac{PdV_1}{d\phi} \right] / m_B \cdot Cv_1 \tag{9}$$

$$\frac{dT_U}{d\phi} = \left[ -\frac{dm_g^{ent}}{d\phi} \left( h_g^{ent} - u_{g,1} \right) + \frac{dm_{sg}^{exit}}{d\phi} \left( h_{sg}^{exit} - u_{sg,1} \right) - \frac{dQ_{W2}}{d\phi} - \frac{PdV_2}{d\phi} \right] / m_U \cdot Cv_2 \tag{10}$$

The instantaneous temperature model for the unburned zone is Equation (10), which have similar parameters to Equation (9), but the unburned area did not include the exothermic combustion. Where $C_v$ is the constant volume specific heat capacity, $h_g^{ent}$, $u_g^{ent}$ are the enthalpy and thermodynamic energy of the gas leaving the unburned zone, respectively, $dQ_{w1}/d\phi$ and $dQ_{w2}/d\phi$ are the heat loss at the cylinder wall for the burned and unburned zones, respectively, $P$ is the in-cylinder pressure, $dV_1$ and $dV_2$ are the instantaneous volumes of the burned and unburned zones, respectively.

### 2.5. Two-Zone Model Volume Balance

In the process of in-cylinder combustion, the volume balance is important to calculate the volume and in-cylinder pressure of each combustion zone, since the volume ratio of the burned and unburned zones is changeable after the combustion starts. Under the assumptions that the gas in the two-zone model conforms to the ideal gas equation of state and that the pressures in both zones are the same, the volume equilibrium in both regions could be solved using the ideal gas equation of state, as follows:

$$V_z = \frac{\pi D^2}{4} \left\{ \frac{S}{\varepsilon - 1} + \frac{S}{2} \left[ 1 + \frac{1}{\lambda} - \cos\left(\frac{\pi\phi}{180}\right) - \sqrt{\frac{1}{\lambda^2} - \sin^2\left(\frac{\pi\phi}{180}\right)} \right] \right\} \tag{11}$$

$$L = \frac{S}{\varepsilon - 1} + \frac{S}{2} \left[ 1 + \frac{1}{\lambda} - \cos\left(\frac{\pi\phi}{180}\right) - \sqrt{\frac{1}{\lambda^2} - \sin^2\left(\frac{\pi\phi}{180}\right)} \right] \tag{12}$$

$$V_1 = \frac{(mRT)_B}{P} \tag{13}$$

$$V_2 = V_z - V_1 \tag{14}$$

$$P = \frac{(mRT)_U}{V_2} \tag{15}$$

Cylinder instantaneous volume $V_z$ is the parameter that varies with the crankshaft rotation angle, calculated from $\phi = 0$ when the crank is at the upper dead center position. In addition, $\varepsilon$ is the compression ratio; $D$ is the cylinder diameter; $S$ is the cylinder stroke; and $\lambda$ is the crank ratio of the connecting rod.

Figure 3 is the flow chart of volume calculation for the dual area model. First, the volume $V_1$ of the burned zone was calculated according to the ideal gas equation. Then, the unburned zone volume $V_2$ was obtained by subtracting the burned zone volume $V_1$ from the instantaneous volume $V_z$ of the cylinder, corresponding to the crankshaft angle. Finally, the corresponding pressure $P$ could be obtained by using the ideal gas equation in the unburned zone. As the pressure of the combustion zone and the unburned zone was the same, the pressure was the engine cylinder pressure $P$.

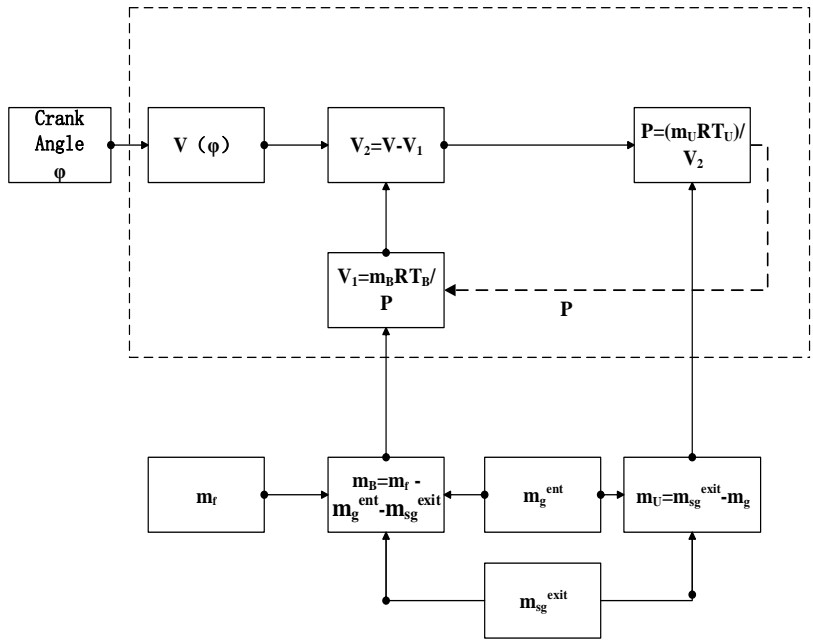

**Figure 3.** Flow chart of volume calculation for the two-zone model.

### 2.6. Heat Transfer Loss

The cylinder wall is composed of the surface of the combustion chamber of the cylinder head, the top surface of the piston, and the instantaneous heat transfer surface area of the cylinder liner. However, in this study, we divided the cylinder into a burned zone and an unburned zone, and the volume of the two zones was added to the total volume of the cylinder. Assuming that the volume of the two areas was in contact with the cylinder wall, both will dissipate heat outside the wall. The heat dissipation through the wall could be calculated by the instantaneous average heat transfer coefficient $\alpha g$ of the working fluid to the cylinder peripheral wall and the average wall temperature $T_w$, as shown in the following equation:

$$\frac{dQ_{wi}}{d\phi} = A_{\text{wall},i} \cdot \alpha_w \cdot (T_i - T_w) \tag{16}$$

$$\alpha_w = 2.06 \cdot 10^{-7} D^{-0.2} P^{0.8} T_i^{-0.53} \left[ C_1 \cdot C_m + C_2 \cdot \frac{V_S T_a}{P_a V_a} (P - P_0) \right]^{0.8} \tag{17}$$

$i$ = 1, 2, respectively, representing the burned area and the unburned area. $A$ is the heat exchange area, calculated according to the geometry of the cylinder. Since the dual-zone model divides the cylinder into two volumes—the burned zone and the unburned zone—the burned zone releases heat to the cylinder head and cylinder wall, while the unburned zone transfers heat release to the piston crown and cylinder wall. The area of the cylinder wall was calculated from the volume of each zone.

$$A_{\text{head}} = A_{\text{crown}} = \frac{\pi}{4} D^2 \tag{18}$$

$$A_{\text{wall}} = \pi \cdot L \cdot D \tag{19}$$

$$A_{\text{wall},2} = S_L \cdot A_{\text{wall}} \tag{20}$$

$$A_{\text{wall},1} = (1 - S_L) \cdot A_{\text{wall}} \tag{21}$$

where $L$ is calculated according to Equation (12), $A_{\text{head}}$ and $A_{\text{crown}}$ are the surface areas of the cylinder head and piston crown, respectively; $A_{\text{wall},1}$ and $A_{\text{wall},2}$ are the heat dissipation area of the cylinder wall in the burned zone and the unburned zone, respectively. $S_L$ is the distribution factor, which is defined as the ratio of the unburned zone volume to the total cylinder volume:

$$S_L = \frac{V_2}{V_z} \tag{22}$$

### 2.7. Combustion Model

In this model, the combustion process was divided into two stages—premixed combustion and diffusion combustion. The premixed combustion stage was represented by $dx_1/d\phi$, and the diffusion combustion stage was represented by $dx_2/d\phi$. The proportion of each stage was controlled by the number of combustion fuel.

$$\frac{dx}{d\phi} = \frac{dx_1}{d\phi} + \frac{dx_2}{d\phi} \tag{23}$$

$$\frac{dx_1}{d\phi} = \left\{ \frac{6.908 \cdot (m_p + 1)}{\phi_{Zp}} \cdot \left( \frac{\phi - \phi_B}{\phi_{Zp}} \right)^{m_p} exp \left[ -6.908 \left( \frac{\phi - \phi_B}{\phi_{Zp}} \right)^{m_p+1} \right] \right\} \cdot (1 - Q_d) \tag{24}$$

$$\frac{dx_2}{d\phi} = \left\{ \frac{a \cdot (m_d + 1)}{\phi_{Zd}} \cdot \left( \frac{\phi - \phi_B}{\phi_{Zd}} \right)^{m_d} exp \left[ -a \left( \frac{\phi - \phi_B}{\phi_{Zd}} \right)^{m_d+1} \right] \right\} \cdot Q_d \tag{25}$$

where the subscript $p$ is the premixed combustion; the subscript $d$ is the diffusion combustion; $m$ is the combustion quality coefficient; $\phi_Z$ is the combustion duration; $\phi_B$ is the combustion advance angle; and $Q$ is the combustion fraction.

*2.8. Knock Model*

In this study, the knock prediction model and the Knock Intensity (*KI*) prediction model were combined as indicators to simulate the time, trend, and intensity of engine knock when the engine parameters changed. The knock prediction model adopted the knock integral model of the end mixture spontaneous combustion delay period integral method, improved by Douaud [18]. In the model of Douaud and Eyzat, knock is predicted to occur at the crank angle at which the induction time integral (for any unburned zone) attains a value of 1.0. At the same time, the effect of octane number on knock was considered in this model:

$$I = \int_{t=0}^{t_i} \frac{dt}{\tau} = 1 \tag{26}$$

In Equation (26), $\tau$ is the induction time calculated from Equation (27), which is a function of instantaneous temperature and pressure in the unburned zone, the unit is milliseconds. $t$ is the time elapsed from the start of the compression process in the unburned zone. $t_i$ is the time when spontaneous combustion occurs.

$$\tau = 5.72 \cdot 10^6 \cdot B \cdot \left(\frac{ON}{100}\right)^{3.402} P^{-1.7} \cdot 10^{\left(\frac{3800}{K \cdot T_U}\right)} \tag{27}$$

In Equation (27), Octane number (ON) is the fuel octane number; $P$ is the instantaneous pressure; $T_U$ is the temperature of the unburned zone; $B$ is the burst-induced time multiplier; $K$ is the activation of energy multipliers (when the value is greater than 1, it increases the likelihood of knock).

The Knock Intensity (*KI*) predictive model is a phenomenology-based parameter that was developed by Gamma Technologies Laboratories to describe knock intensity and loudness. The knock index is defined as a crank angle-dependent quantity as follows:

$$KI = 10000M \cdot u \frac{V_{TDC}}{V_Z} exp(\frac{-6000}{T_U})\max[0, 1 - (1 - \phi)^2] \cdot I \tag{28}$$

Where $M$ is the knock index multiplier; $u$ is the percentage of unburned zone mass in the cylinder; and $\phi$ is the equivalence ratio of the unburned zone. When knock occurs, the value of the knock index at the start of knock is reported. Calibrating the parameters of the knock model makes *KI* equal to 200 and the slight knock of engine in the corresponding experiment. In other words, when the *KI* value is larger than 200, it is considered that the knock occurs, and the higher the *KI* value, the higher is the knock intensity.

## 3. Model Calibration and Validation

In this paper, the MAN 8L51/60DF engine is used as the prototype of the model, all data of engine modeling and calibration are derived from the shipyard's factory test reports certified by China Classification Society and Lloyd's Register of Shipping. The model calibration could also be found in the above research [19–21]. The engine layout and components are presented in Figure 4 [9]. The main engine parameters are listed in Table 2. Figure 5 is the established two-zone simulation model of the MAN 8L51/60DF engine.

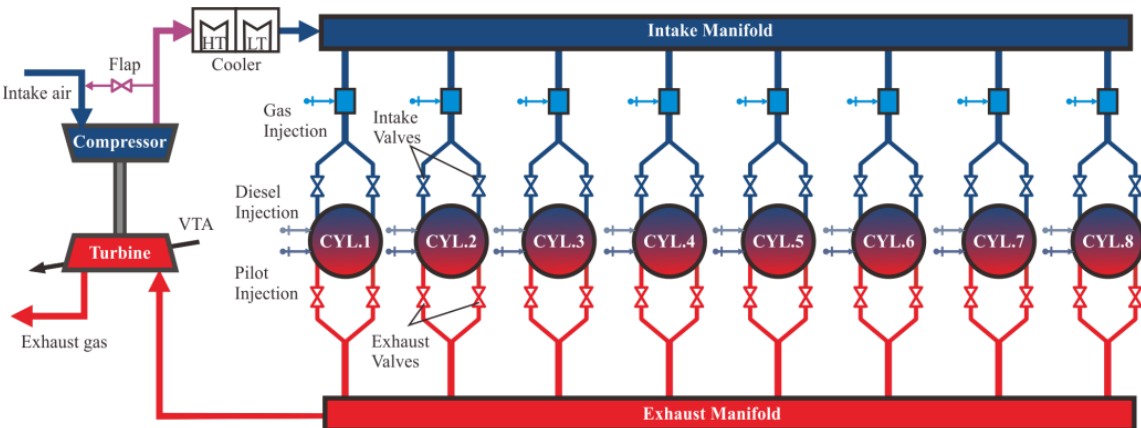

**Figure 4.** Engine layout and components [9].

**Table 2.** Engine technical parameters.

| Parameter | Unit | Values |
|---|---|---|
| Cylinder number | [-] | 8 |
| Bore | [mm] | 510 |
| Stroke | [mm] | 600 |
| Compression ratio | [-] | 13.3 |
| Power | [kW] | 8000 |
| Rated speed | [r/min] | 514 |
| Fire order | [-] | 1-4-7-6-8-5-2-3 |
| IMEP | [bar] | 19.1 |

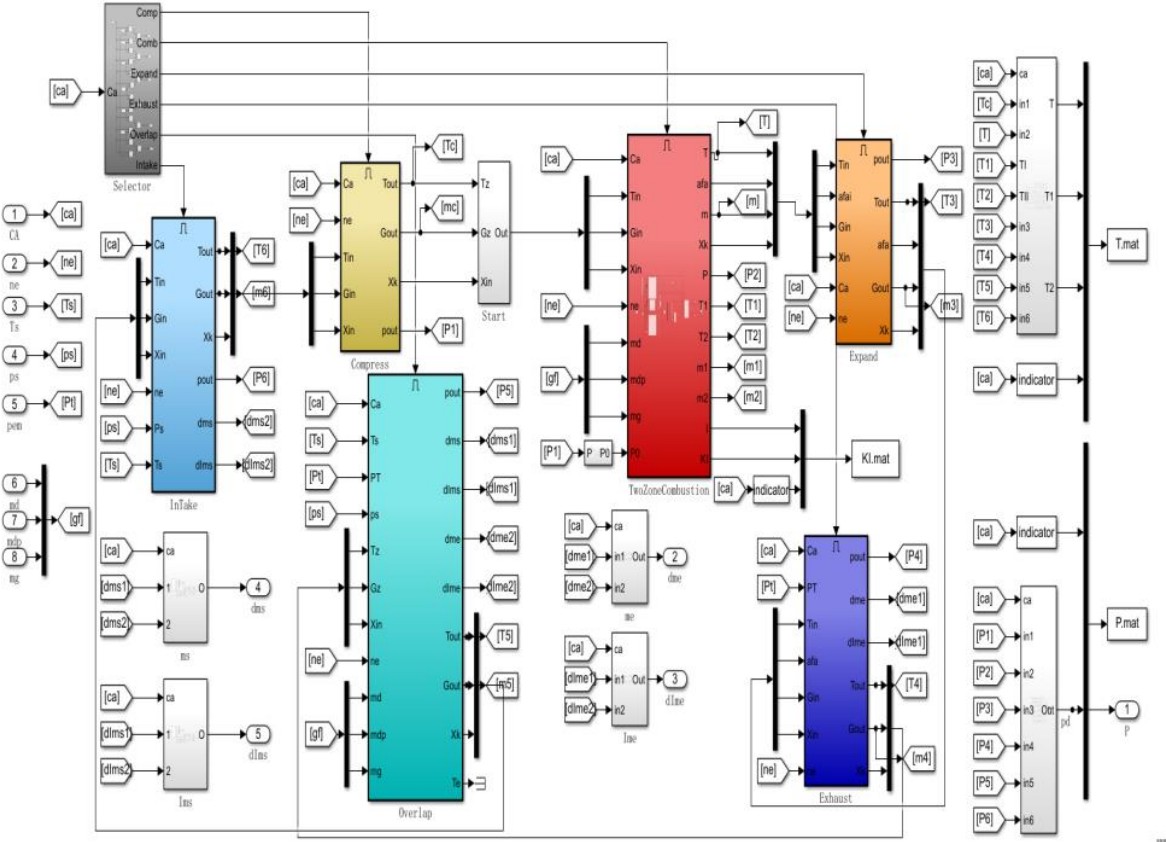

**Figure 5.** Two-zone simulation model.

The boundary, initial and operation conditions for the numerical simulation are illustrated in Table 3. Figure 6 shows the comparison between the cylinder pressure simulated by the quasi-dimensional model and the engine experiment pressure, and the cylinder pressure curve of Gan et al. [20] was used to validate the quasi-dimensional simulation model. Figures 7–9, respectively, show the comparison between the simulated and experimental values of the Max.Firing.Pressure, engine power, and Indicated Mean Effective Pressure (IMEP), under different engine loads. It could be seen from the figure that the pressure in the cylinder was very accurate. The relative errors of the peak pressure between the cylinder pressure and the test pressure under each load were very small. The simulation was consistent with the experimental trend, and the parameters fit very well under each load. Figure 10 is the simulation curve of the heat release rate of cylinder. The whole heat release rate curve truly reflected the process of pre-combustion and diffusion combustion. It could be seen from the simulation results that the model simulation accuracy were relatively high, which could be used to study the knock performance simulation of dual-fuel engines.

**Table 3.** The boundary, initial, and operation conditions for the model.

| Parameter | Unit | Values |
|---|---|---|
| Operation conditions | | |
| Load | [-] | 100% |
| Fire order | [-] | 1-4-7-6-8-5-2-3 |
| Rated speed | [r/min] | 514 |
| SOI | [°ATDC] | −15 |
| Initial conditions | | |
| Intake temperature | [K] | 316 |
| Intake pressure | [bar] | 4 |
| Natural gas intake flow | [kg/cycle] | 0.01025 |
| Exhaust temperature | [K] | 777 |
| Exhaust pressure | [bar] | 3 |
| Boundary conditions | | |
| Piston crown temperature | [K] | 523 |
| Cylinder head temperature | [K] | 553 |
| Cylinder wall temperature | [K] | 433 |

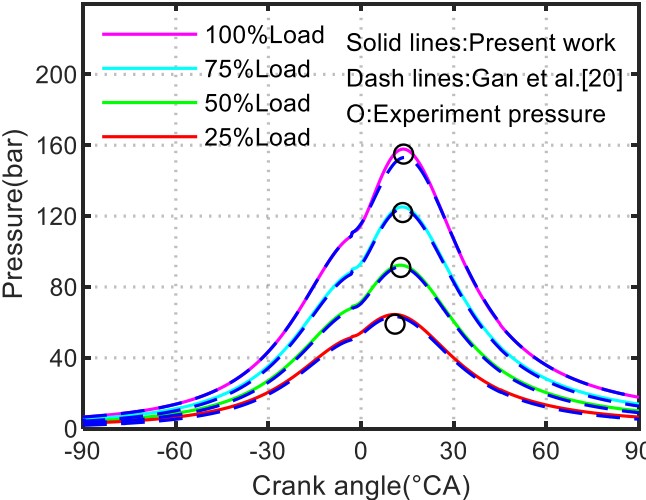

**Figure 6.** Cylinder gas pressure.

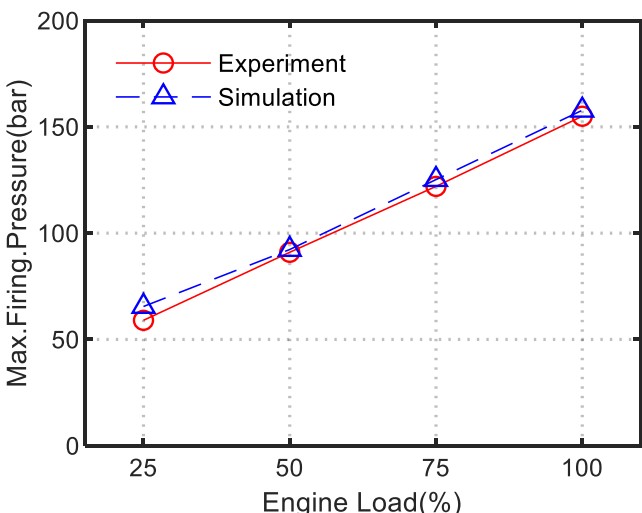

**Figure 7.** Max.Firing.Pressure comparison between simulation and experiment.

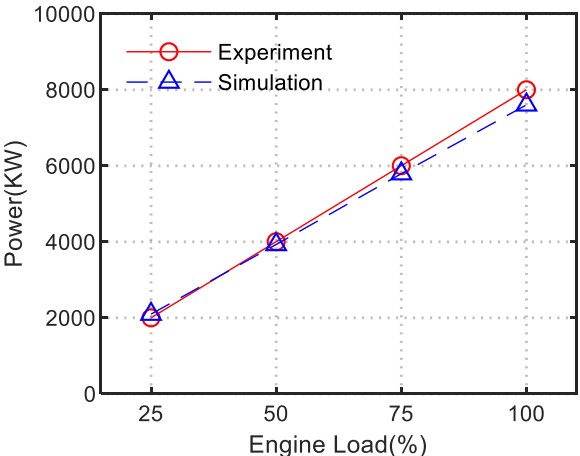

**Figure 8.** Power comparison between simulation and experiment.

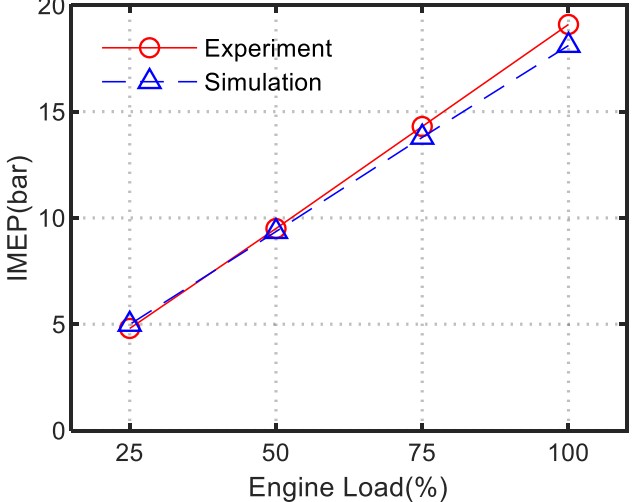

**Figure 9.** IMEP comparison between simulation and experiment.

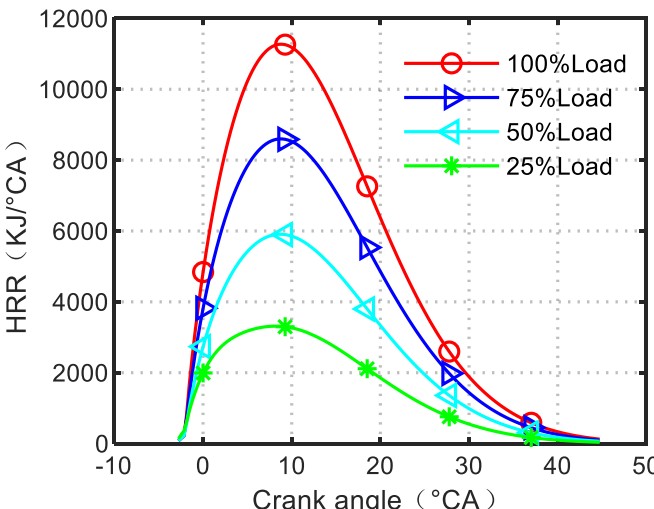

**Figure 10.** Cylinder heat release rate simulation curve.

Figure 11 compares the in-cylinder temperatures of the two-zone model and the zero-dimensional model. The instantaneous temperature of the burned zone and the unburned zone can be calculated by Equations (9) and (10). It could be seen that the temperatures of these two models remained constant during the compression and expansion phases. The temperature curves began to separate at the beginning of combustion and coincided again at the end of combustion. In the separated combustion stage, the peak temperatures of the burned and the unburned zones reached 2411 K and 1203 K, respectively, while the peak temperature of the zero-dimensional model was 1737 K. This was consistent with the results of the two-zone model of direct injection engine by Galindo Lopez [22] of Delft University in the Netherlands. This also showed the accuracy of the simulation model to a certain extent.

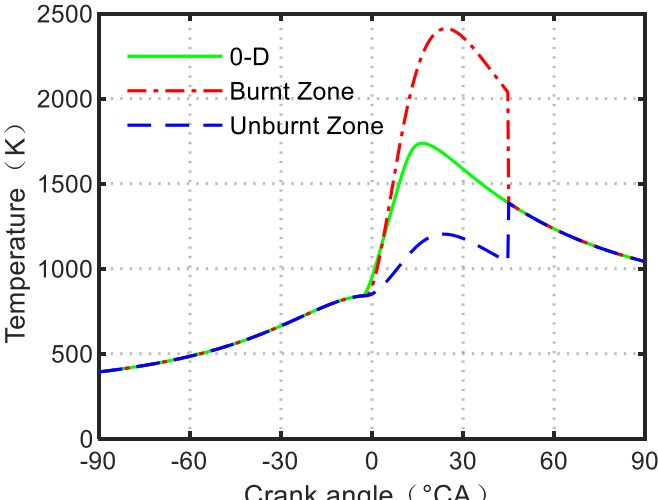

**Figure 11.** Gas temperature in cylinder.

## 4. Analysis of Simulation Results

The accuracy of the model was verified by calibrating the experimental data of the MAN 8L51/60DF dual-fuel engine, under various load conditions. In order to further study the knock characteristics of the marine dual-fuel engines, on the basis of the established two-zone simulation model, this paper used the knock simulation model to set three groups of engine parameters, engine intake temperature, compression ratio, and natural gas intake as variables. When one of the above parameters is taken as a variable, the other parameters

are controlled unchanged. The simulation results of the model could analyze the effect of changing engine parameters on the timing of knocking and the intensity of knock.

### 4.1. Effect of Natural Gas Intake on Knock

The main fuel of the dual-fuel engine studied in this paper is natural gas, so it has a great influence on the combustion process in cylinder. Natural gas and air enter the cylinder after being mixed at the intake end, and are ignited by the combustion pilot diesel near the top dead center of the piston. The knock of the engine mainly occurs during the natural gas combustion, so the natural combustion has a great impact on the knock combustion characteristics of the engine. In this section, the in-cylinder injection timing, intake temperature, and geometric compression ratio are set as the fixed values, and the natural gas quality in the initial combustion chamber is taken as the variable to study the influence of different intake volumes of natural gas on the knock combustion characteristics of the engine. Table 4 shows the simulation parameters under different natural gas intake conditions.

**Table 4.** The simulation parameters.

|  | $m_1$ = 9.25 g | $m_2$ = 10.25 g | $m_3$ = 11.25 g | $m_4$ = 12.25 g |
|---|---|---|---|---|
| Natural gas intake | 9.25 g | 10.25 g | 11.25 g | 12.25 g |
| Rated speed [r/min] | 514 | 514 | 514 | 514 |
| Intake temperature [K] | 316 | 316 | 316 | 316 |
| Compression ratio | 13.3 | 13.3 | 13.3 | 13.3 |
| SOI [°ATDC] | −15 | −15 | −15 | −15 |

The effect of changing the natural gas intake on the knock integral is shown in Figure 12. With an increase in the quality of natural gas in the combustion chamber, the knock integral increased rapidly, and the knock phenomenon occurred. The timing of knock also advanced with an increase in natural gas intake, and the knock intensity also increased rapidly, as shown in Figure 13.

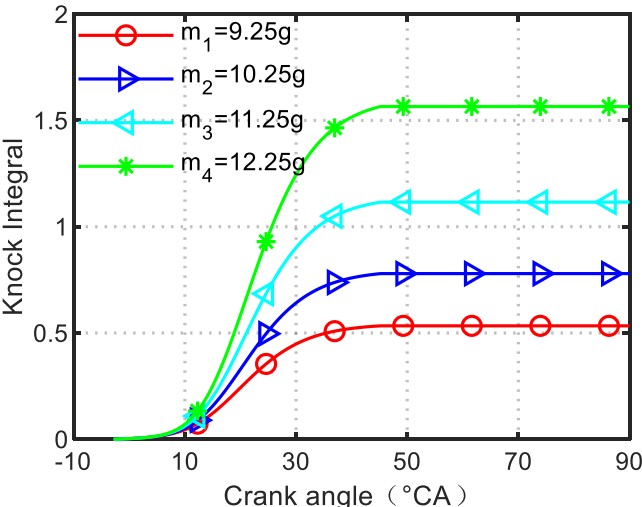

**Figure 12.** Knock integral under different natural gas intake.

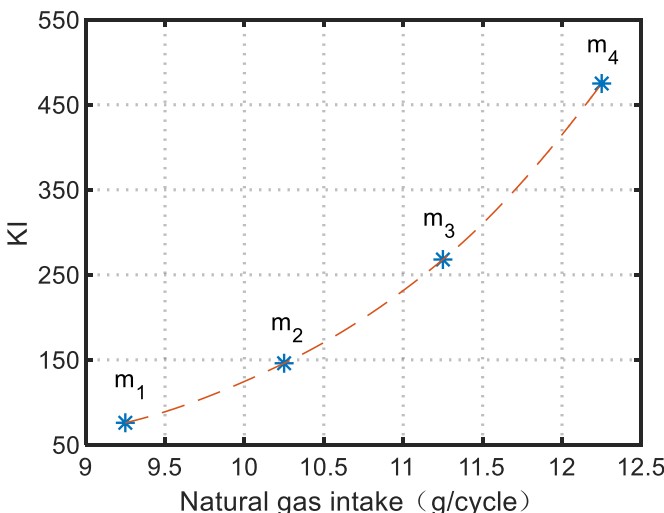

**Figure 13.** Knock intensity under different natural gas intake.

It can be seen that the quality of natural gas has a significant impact on the time and intensity of engine knock. The more natural gas intake, the more fuel involved in the combustion in the combustion chamber, the more intense the combustion, and the greater the engine knock intensity. On the contrary, the less natural gas intake, the smaller the engine knock intensity.

At the same time, the amount of natural gas intake also has an effect on the time of engine knock. The larger the amount of natural gas intake, the earlier the engine knock occurs. Therefore, in order to avoid the occurrence of knock, the intake of natural gas should be reasonably controlled.

### 4.2. Effect of Intake Temperature on Knock

In the process of research, the average temperature in cylinder at the beginning of combustion simulation is taken as the intake temperature, and the rotational speed, compression ratio, injection timing, and natural gas intake volume are kept unchanged. Pure methane is used as the fuel for combustion simulation. The intake temperature is set to 296 K, 316 K, 336 K, and 356 K, respectively, to simulate and analyze the influence of four different intake temperatures on the engine knock phenomenon. Table 5 shows the simulation parameters under four different intake temperature conditions.

**Table 5.** The simulation parameters.

|  | $T_1$ = 296 K | $T_2$ = 316 K | $T_3$ = 336 K | $T_4$ = 356 K |
|---|---|---|---|---|
| Natural gas intake | 10.25 g | 10.25 g | 10.25 g | 10.25 g |
| Rated speed [r/min] | 514 | 514 | 514 | 514 |
| Intake temperature [K] | 296 | 316 | 336 | 356 |
| Compression ratio | 13.3 | 13.3 | 13.3 | 13.3 |
| SOI [°ATDC] | −15 | −15 | −15 | −15 |

The intake temperature of the engine cylinder was set to 296 K, 316 K, 336 K, and 356 K, respectively. Figure 14 shows the knock integral curve at different intake temperatures. Figure 15 is the knock intensity index coefficient. As the intake air temperature increased, the knock intensity index coefficient increased rapidly, this was the same trend as shown by the knock integral curve in Figure 14.

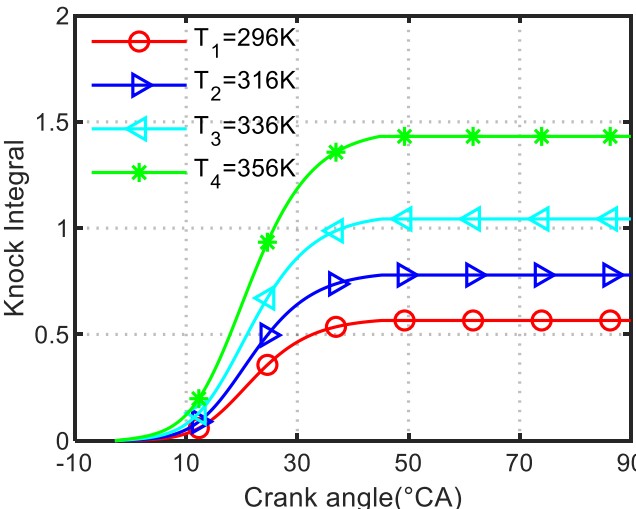

**Figure 14.** Knock integral under different intake temperature.

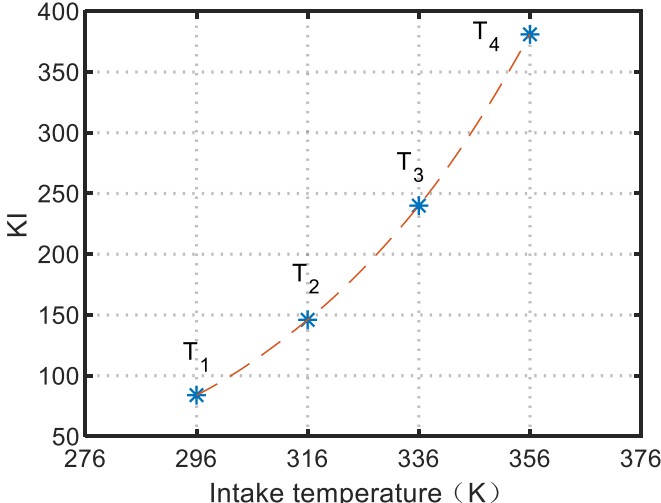

**Figure 15.** Knock intensity at different intake temperatures.

The results showed that the higher the intake temperature, the higher the temperature of the unburned zone in the cylinder. Therefore, affected by the intake temperature of the engine cylinder, the higher the intake temperature, the greater the possibility of engine knock, and the greater the knock intensity. Controlling the intake temperature of the engine and reducing the thermal load in the engine cylinder could reduce the occurrence of the engine knock.

### 4.3. Effect of Compression Ratio on Knock

The faster the natural gas engine combustion fuel with a larger compression ratio, the greater the power output, and the better the economic benefit. However, the valve was not closed tightly due to the wear and ablation during engine operation, which would cause abnormal engine compression ratio. The larger the compression ratio, the more the gas was compressed, and the higher the temperature and pressure rise. However, the larger the compression ratio, the higher the pressure and temperature in the cylinder. Before the flame front arrives, the combustion was more likely to ignite spontaneously. Therefore, when the compression ratio reached a certain limit, it would also be accompanied by the possibility of a knock. In this section, the two zone combustion model would be used to simulate the influence of compression ratio change on knock. Table 6 shows the simulation parameters under different compression ratio conditions.

**Table 6.** The simulation parameters.

| | $\varepsilon_1 = 11.3$ | $\varepsilon_2 = 13.3$ | $\varepsilon_3 = 15.3$ | $\varepsilon_4 = 17.3$ |
|---|---|---|---|---|
| Natural gas intake | 10.25 g | 10.25 g | 10.25 g | 10.25 g |
| Rated speed [r/min] | 514 | 514 | 514 | 514 |
| Intake temperature [K] | 316 | 316 | 316 | 316 |
| Compression ratio | 11.3 | 13.3 | 15.3 | 17.3 |
| SOI [°ATDC] | −15 | −15 | −15 | −15 |

The compression ratio of the engine $\varepsilon$ was set as $\varepsilon_1 = 11.3$, $\varepsilon_2 = 13.3$, $\varepsilon_3 = 15.3$, and $\varepsilon_4 = 17.3$, respectively. Figure 16 shows the knock integral curve at different compression ratios. The knock intensity is shown in Figure 17. It could be seen from the figure that as the compression ratio increased, the knock intensity increased. This was because the higher the compression ratio, the higher was the cylinder pressure and the unburned zone temperature. Before the flame front arrived, the combustion was more prone to spontaneous combustion. It could be seen from the simulation results in Figure 16 that the knocking phenomenon was affected by the engine compression ratio. The larger the compression ratio, the earlier the engine knock occurred. The advance of the knock time would cause more natural gas to spontaneously ignite in the unburned zone, release more energy, and result in a stronger knock intensity. The reasonable compression ratio of the engine could reduce the probability of a knock.

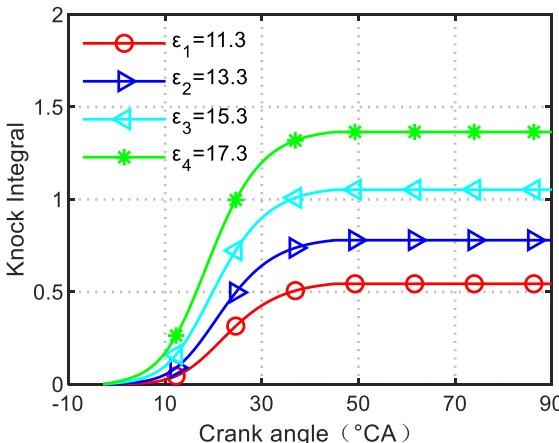

**Figure 16.** Knock integral under different compression ratio.

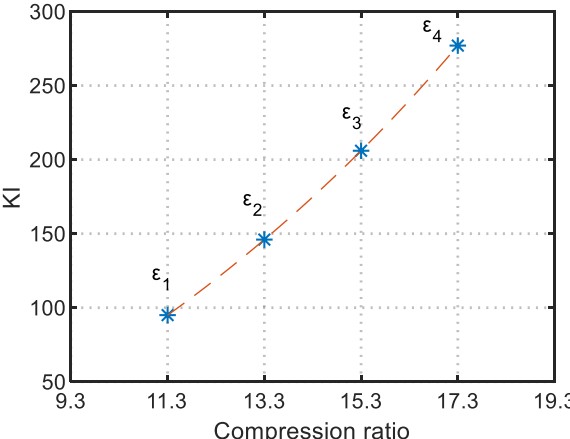

**Figure 17.** Knock intensity under different compression ratio.

## 5. Conclusions

This paper takes the MAN 8L51/60DF dual-fuel engine as the research object, focusing on the establishment of dual zone simulation model and knock simulation of the marine dual-fuel engine research. In order to ensure the reliability of the simulation analysis in this paper, the model was verified by bench test data, and the simulation results fit well, which showed that the established model was suitable for a dual-fuel engine knock simulation research. Three parameters of natural gas intake, intake temperature, and compression ratio were selected as variables to simulate the engine. Through the analysis of simulation results, the following conclusions were drawn:

(1) The intake of natural gas was $m_1 = 9.25$ g, $m_2 = 10.25$ g, $m_3 = 11.25$ g, and $m_4 = 12.25$ g. According to analysis of the simulation results, during engine combustion, the more natural gas intake, the more fuel involved in combustion. Natural gas burns to release more heat, the more intense the combustion, the greater the probability of engine knock. The more natural gas intake, the earlier the engine knock occurs, and the greater is the knock intensity. Therefore, the amount of natural gas intake has a significant impact on the engine knock phenomenon. Reasonable control of the intake of natural gas is of great significance to reducing engine knock.

(2) The intake temperature of the engine cylinder was set to $T_1 = 296$ K, $T_2 = 316$ K, $T_3 = 336$ K, and $T_4 = 356$ K respectively. The higher the intake temperature, the higher the temperature of the unburned zone in the cylinder. Therefore, affected by the intake temperature of engine cylinder, the higher the intake temperature, the greater the possibility of an engine knock and the greater the knock intensity. By controlling the intake temperature of the engine and reducing the thermal load in the engine cylinder, the occurrence of engine knock could be reduced.

(3) The engine compression ratio was set to $\varepsilon_1 = 11.3$, $\varepsilon_2 = 13.3$, $\varepsilon_3 = 15.3$, and $\varepsilon_4 = 17.3$. The larger the compression ratio, the higher the cylinder pressure and the unburned zone temperature. Before the flame front arrived, the combustion was more likely to ignite spontaneously. The simulation results showed that the knock phenomenon was affected by the compression ratio of the engine. The higher the compression ratio, the earlier the engine knock occurred and the greater the knock intensity. The reasonable compression ratio of the engine could reduce the probability of a knock.

**Author Contributions:** F.-k.Z.: Conceptualization, Data curation, Investigation, Methodology, Software, Validation, Visualization, Writing—original draft; H.Z.: Corresponding author, Project administration, Resources, Validation, Writing—review & editing, Funding acquisition; H.-y.W.: Conceptualization, Supervision, Methodology, Validation, Writing—review & editing; X.-x.W.: Formal analysis, Supervision; Z.-x.X.: Validation. All authors have read and agreed to the published version of the manuscript.

**Funding:** This research was funded by Key Technology Research of Marine Equipment Intelligent Integration and Remote Management, grant number [CJ02N20], and the Fundamental Research Funds for the Central Universities, grant number [No.3132016316].

**Data Availability Statement:** All data used to support the findings of this study are included within the article.

**Conflicts of Interest:** The authors declare no conflict of interest.

## Nomenclature

| | |
|---|---|
| $A$ | Heat exchange area, m$^2$; |
| $A_{crown}$ | Piston top surface area, m$^2$; |
| $A_{head}$ | Cylinder head surface area, m$^2$; |
| $A_{wall,1}$ | Heat dissipation area of cylinder wall in burned zone, m$^2$; |
| $A_{wall,2}$ | Heat dissipation area of cylinder wall in unburnt zone, m$^2$; |
| $c_{pk}$ | Specific heat at constant pressure, J/(kg·K); |
| CYL | Cylinder |
| HRR | Heat release rate, J/°CA |
| $h_a^{ent}$ | Specific enthalpy of air entering the burned zone, J/kg; |
| $h_{sg}^{ent}$ | Specific enthalpy of chemical reaction gas entering the burned zone, J/kg; |
| $h_{sg}^{exit}$ | The specific enthalpy of the chemical reaction gas entering the unburned zone from the burned zone, J/kg; |
| $I$ | Knock integral; |
| $KI$ | Knock intensity; |
| $m_B$ | Gas mass in the burned area in the cylinder, kg; |
| $m_U$ | The mass of unburned gas in the cylinder, kg; |
| $m_a^{ent}$ | Air quality flowing from unburned area to burned area, kg; |
| $m_f$ | The mass of fuel injected into the cylinder, kg; |
| $m_g^{ent}$ | The mass of gas entering the burned zone in the unburned zone, kg; |
| $m_{sg}^{ent}$ | The mass of the chemical reaction gas entering the burned zone in the unburned zone, kg; |
| $m_{sg}^{exit}$ | The mass of the chemical reaction gas flowing from the burned area to the unburned area, kg; |
| $m_z$ | The total mass of the substance in the cylinder, kg; |
| $P$ | Gas pressure in the cylinder, pa; |
| $Q_f$ | The heat brought by the fuel injected into the cylinder, J; |
| $Q_{w1}$ | Heat taken away by the cooling medium in the burned area, J; |
| $Q_{w2}$ | Heat taken away by the cooling medium in the unburned area, J; |
| $S_L$ | Distribution factor, the ratio of the unburned zone volume to the nstantaneous total volume of the cylinder; |
| SOI | Start of Injection |
| $T_B$ | Instantaneous temperature in the burned area, K; |
| $T_U$ | Instantaneous temperature of unburned zone, K; |
| VTA | Variable Turbine Area |
| $V_1$ | Instantaneous volume of burned area, m$^3$; |
| $V_2$ | Instantaneous volume of unburned area, m$^3$; |
| $V_z$ | The instantaneous volume of the cylinder, m$^3$. |

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
