# Peer review of "Implementation and Parameter Analysis of the Knock Phenomenon of a Marine Dual-Fuel Engine Based on a Two-Zone Combustion Model"

_processes, doi:10.3390/pr9040602_

Round 1

Reviewer 1 Report

1) As the dual-fuel engines and their analysis/simulations are recently important research topics, in the Introduction should be presented a better and wider overview of already published papers in this research field. In the literature can be found many recently published papers related to dual-fuel engines and its processes, therefore an Introduction require a better overview of already obtained conclusions in this research field. Also, used papers are dominantly older than 5 years.

2) In the paper should be added a Nomenclature in which all the abbreviations, marking, symbols, etc. should be explained. Some of them are not explained in the paper at the moment, so the Nomenclature will resolve this problem.

3) The whole paper should be carefully checked from the English viewpoint. In several places occurs typing or spelling errors (for example, under Eq. 10 is written – Whree, etc.). All such obvious mistakes should be removed.

4) Section 2 - Model Description. The Authors show standard equations for 0D and 2D numerical models. Is there any improvement or novelty inside this numerical model which is not already known and which cannot be found in the literature? In the literature can be also found several complex models in which 0D model is integrated with QD model (QD model is used for combustion process). Such models are well developed for single-fuel as well as for dual-fuel engines. In this paper the Authors integrated 0D model with 2D model for combustion. As such integration is not novel – I expected some kind of novelty in the numerical model. As presented, I cannot find any novelty in this integration.

5) Section 3 - Model calibration and validation – as the Authors stated, the numerical model was verified under 100% load conditions. There is no such numerical model which can be considered as a reliable, accurate and precise if the validation is not performed by using at least several engine loads. Validation by using only one engine load is insufficient, so the validation by using some other engine loads must be performed.

6) In relation to my previous comment (comment 5), validation shown in Table 2 cannot be considered as adequate (at least in my own opinion). Firstly, all of the presented results are related to only one load. Secondly, some of the variables should not be presented only as a one value. For example - Average indicated pressure will have small relative error also with numerical models which performance strongly differs in relation to measurements. The average values can hide many problems which can exist in the numerical model. Therefore, such kind of validation did not prove that the numerical model is reliable, accurate and precise. Proper validation will be to show (for example) the entire change of cylinder indicated pressure and heat release rate in the cylinder (measured values) and direct comparison with the same variables calculated by using numerical model (the entire curves, not only one or average point) for two or more engine loads. Finally, validation presented in this paper is not, in my opinion, proper and adequate.

7) Figure 5 should be presented for various loads (as well as heat release rate). Also, it is presented a measured curve. How this measured curve is obtained? With which measurement equipment? On which engine cylinder? When presenting measurement results, it is expected to describe a measurement procedure, give an information about measuring equipment and the most important details about measurements. The change of in-cylinder pressure is the only measured variable shown in this paper. I have never seen engine measurements where the only measured parameter is a change of in-cylinder pressure at only one engine load, especially without describing measurement procedure and equipment. More measurement results should be presented, especially for the validation purposes.

8) Figure 10 legend is wrong (it is missing one curve, it should be presented injection time, not compression ratios).

9) It is not clear what the scientific novelty of this paper is. Both used numerical models are already known (connected together as well as independently) and the Authors did not improve any of them. Measurement procedure and validation are inadequate (or non-existing). Shown variations (Section 4) can be obtained with many different numerical models. Therefore, I did not see any novelty presented in this paper, which cannot be already found in the literature.

Reviewer 2 Report

Manuscript "Research on Knock Simulation of Dual Fuel Engine Based on Two-Zone
Model" by ZOU Fang-kun, ZENG Hong, ZHANG Jin-hao1 and WANG Huai-yu is belonged
to very-very applied science. Certainly engineers need the trusted tool to
predict a knock in gas engines. Unfortunately the manuscript is written on such
poor English, that some times I can not understand the sence of the statement.
Moreover there are some statements where it is not clear which word is subject,
and which one is predicate.
For example:
"In order to analyze the causes of the burst phenomenon and the influence of
operating parameters on knock." (Page 1, line 5). The predicate is omitted
here.
Another example:
" Another, the 3D simulation model provides most of the details of the
in-cylinder parameters are usually complex and extremely computationally
intensive,..." (Page 1, line 8 from the bottom). There are the two predicates
("provides" and "are") in the same statement.
At this stage I recommend to authors rewrite the text using the simple
statements only, i.e. one subject and one predicate have to be in one statement.

Some other comments:

1. Authors solved ODE (1)-(4). Which initial conditions they employ?

2. Assumption (4) "The combustion products of natural gas and air include
only CO2, H2O, O2, N2 and Ar;" Where and how does authors use the assumption?

3. Q_f - the heat of combustion (?). How authors calculate it?

4. How authors calculate Q_w1 and Q_w2?

5. How authors estimate B and A in eq. (10).

6. Fig. 10. What is plotted in black?

Reviewer 3 Report

Zou and colleagues used a two-zone model of a dual-fuel engine simulation model, developed in MATLAB/Simulink scientific tool, to investigate a so-called knock phenomenon. They used a zero-dimensional model with a two-zone model. They introduced the model divided into six (6) modules according to the in-cylinder mass changes. The flame front divides the combustion chamber into two zones: burned and unburned. Further, the temperature in the unburned zone has been used as an input variable to determine burst prediction.

In my opinion, the investigated subject has great potential to be published. The research topic is actual and worth exploring.

However, I have some reservations about the manuscript in the current form. In my opinion, the manuscript is rushed, i.e., it is half-finished.

First, I think the manuscript is hard to read. The interpunction sings are missing, and some sentences are combined. Therefore, it is hard to follow the authors’ thoughts, which degrades interpretation clarity and quality. I advise the authors to revise the text again.

Major concerns:

  1. What is your original contribution to the manuscript? From the text, it is not clear, whether, did you develop the simulation model, i.e., a dual-fuel engine two-zone model? Please, can you elaborate in more detail?
  2. In conclusion, you wrote: “(1) The two zone model has high accuracy in predicting engine performance and knock simulation, and the calculation time cost and complexity of the two zone model are much less than those of the three-dimensional model.” Please, can you show the proof? Did you compare your model with other existing models and show performances (a calculation time, complexity, etc.) between models?
  3. In the second section, sub-section 2.1. Two-zone model you made five (5) assumptions. For the first four (4), please, can you elaborate, why they were made as they are?

Round 2

Reviewer 1 Report

The Authors have notably improved this paper. However, there are still elements of major importance which must be added into the paper:

1) Model validation - It is necessary to add Heat Release Rate validation. Average indicated pressure (at one or more different loads) will have small relative error also with numerical models which performance strongly differs in relation to measurements.

2) Description of measurement procedure and at least a general specifications of measuring equipment is still completely missing in the paper.

3) I still cannot find proper scientific novelty. If exists, it should be much better described and discussed.

4) The References list - newly added researches are also dominantly older than 5 years. The References list still has to be extended. Without a new, recent researches in this research field - the paper relevance is questionable (or non existing).

Reviewer 2 Report

Authors did not improve the English writing. It is still absolutely not
readable.

1) How "the assumptions" may be "initial conditions"?

2) O.k.

3) What is "double Vibe combustion model"? Where a reader can find it?

4) O.k.

5) O.k.

6) O.k.

Reviewer 3 Report

Thank you very much for addressing the majority of my concerns. The article is fine with me now.

However,  I still have one minor concern, that has not been properly addressed. I urge the authors that at the end of the introduction section, rewrite the aims and clearly state the novelty and scientific contribution of the research topic.

Round 3

Reviewer 1 Report

The Authors have performed corrections. From the previous paper versions, I have understood that the Authors performed its own measurements. As the data is taken from the engine producer specifications - it is impossible to add measuring equipment. So that comment can be neglected.

I believe that now the paper is suitable for publication.

Author Response

Thank you very much for all your suggestions.

Reviewer 2 Report

1) It seems that authors do not understand the term "initial conditions".

Please see https://en.wikipedia.org/wiki/Initial_condition

Author Response

Thank you very much for all your suggestions.Please see the attachment.
